# Renoprotective Effect of the Histone Deacetylase Inhibitor CG200745 in DOCA-Salt Hypertensive Rats

**DOI:** 10.3390/ijms20030508

**Published:** 2019-01-25

**Authors:** Eun Hui Bae, In Jin Kim, Ji Hong Song, Hong Sang Choi, Chang Seong Kim, Gwang Hyeon Eom, Inkyeom Kim, Hyunju Cha, Joong Myung Cho, Seong Kwon Ma, Soo Wan Kim

**Affiliations:** 1Department of Internal Medicine, Chonnam National University Medical School, Gwangju 61469, Korea; remon9127@hanmail.net (I.J.K.); xlzlzl1103@gmail.com (J.H.S.); hongsang38@hanmail.net (H.S.C.); laminion@hanmail.net (C.S.K.); drmsk@hanmail.net (S.K.M.); 2Department of Pharmacology, Chonnam National University Medical School, Hwasun 58128, Korea; eomgh@jnu.ac.kr; 3Department of Pharmacology, School of Medicine, Kyungpook National University, Daegu 41944, Korea; inkim@knu.ac.kr; 4CrystalGenomics, Inc., 5F, Bldg A, Korea Bio Park, Seongnam 13488, Korea; hjcha@cgxinc.com (H.C.); jmcho@cgxinc.com (J.M.C.)

**Keywords:** CG200745, HDAC inhibitor, DOCA, hypertension, fibrosis, inflammation

## Abstract

The novel histone deacetylase inhibitor CG200745 was initially developed to treat various hematological and solid cancers. We investigated the molecular mechanisms associated with the renoprotective effects of CG200745 using deoxycorticosterone acetate (DOCA)-salt hypertensive (DSH) rats. DOCA strips (200 mg/kg) were implanted into rats one week after unilateral nephrectomy. Two weeks after DOCA implantation, DSH rats were randomly divided into two groups that received either physiological saline or CG200745 (5 mg/kg/day) for another two weeks. The extent of glomerulosclerosis and tubulointerstitial fibrosis was determined by Masson’s trichrome staining. The renal expression of fibrosis and inflammatory markers was detected by semiquantitative immunoblotting, a polymerase chain reaction, and immunohistochemistry. Pathological signs such as glomerulosclerosis, tubulointerstitial fibrosis, increased systolic blood pressure, decreased creatinine clearance, and increased albumin-to-creatinine ratios in DSH rats were alleviated by CG200745 treatment compared to those manifestations in positive control animals. Furthermore, this treatment counteracted the increased expression of αSMA, TGF-β1, and Bax, and the decreased expression of Bcl-2 in the kidneys of DSH rats. It also attenuated the increase in the number of apoptotic cells in DSH rats. Thus, CG200745 can effectively prevent the progression of renal injury in DSH rats by exerting anti-inflammatory, anti-fibrotic, and anti-apoptotic effects.

## 1. Introduction

The incidence of chronic kidney diseases (CKD) is increasing worldwide. Hence, therapies capable of curing or inhibiting disease progression are urgently needed. CKD pathogenesis involves fibrotic manifestations, such as glomerulosclerosis and tubulointerstitial fibrosis [1]. The current treatment for CKD is limited to angiotensin-converting enzyme inhibitors and angiotensin receptor blockers, but emerging clinical and experimental evidence indicates that histone deacetylase inhibitors (HDACis) might be candidate therapeutic drugs for CKD. These drugs exhibit anti-fibrotic, anti-inflammatory, anti-hypertrophic, and anti-hypertensive effects [2,3]. Further, epigenetic modifications, such as DNA methylation or histone acetylation, are regarded as important steps in the development and progression of acute kidney injury to CKD. Therefore, epigenetic changes associated with kidney injury and therapeutic targets have been intensively studied [4].

Deoxycorticosterone acetate (DOCA)-salt hypertensive (DSH) rats are a well-established model of mineralocorticoid hypertension with renal dysfunction. Although mineralocorticoids are conventionally thought to promote sodium retention, they also cause oxidative stress [5] and stimulate inflammation and fibrosis [6]. Accordingly, glomerulosclerosis, tubular fibrosis, and cardiac hypertrophy are commonly observed in DSH rats, along with the activation of renal transforming growth factor-β (TGF-β) expression [7]. Therefore, in this study, we sought to investigate whether CG200745 ((E)-2-(naphthalene-1-yloxymethyl)-oct-2-enedioic acid 1-[(3-dimethyl amino-propyl)-amide] 8-hydroxyamide with 1/2(l-tartaric acid)), a recently developed pan-HDACi, has renoprotective effects in the DSH rat model of kidney injury.

## 2. Results

### 2.1. Blood Pressure and Renal Function

Table 1 summarizes the changes in body weight, kidney and left ventricle weight-to-body weight ratio, systolic blood pressure, and urine output in control, DSH, and CG200745-treated DSH rats (DSH + CG200745). Body weight was decreased and the left ventricle weight-to-body weight ratio was increased in DSH rats. Moreover, systolic blood pressure was markedly increased in DSH rats compared to that in control rats, and this difference was attenuated by CG200745 treatment. The left ventricle weight-to-body weight ratio was also decreased by CG200745 treatment, as compared to this value in the DSH group.

Table 2 summarizes renal function changes among the groups. Serum creatinine tended to be increased in DHS rats, but this effect did not reach statistical significance. Further, the fractional excretion of sodium was significantly elevated in DSH rats, suggesting impaired tubular sodium reabsorption. In addition, the albumin-to-creatinine ratio was markedly increased in DSH rats, which was attenuated by CG200745 treatment.

### 2.2. Effect of CG200745 on Acetylation in DSH Rats

Figure 1 shows western blot images of acetyl H3 and acetyl H4 in the different groups. Acetylation was markedly decreased in DSH rats. Treatment with CG200745 prevented the decrease in acetyl H3 in DSH rats, whereas the level of acetyl H4 was even slightly higher than that in control animals. These observations confirmed the efficacy of CG200745 as a histone deacetylase inhibitor.

### 2.3. Effect of CG200745 on Morphological Changes in DSH Rats

Figure 2 shows morphological changes in the kidney cortex in the three groups of experimental animals. Hematoxylin and eosin staining revealed tubular casts, obstruction, and dilatation in DSH rats. In addition, glomerulosclerosis and interstitial expansion were also prominent features of renal injury in DSH rats. Remarkably, all of these changes were attenuated by CG200745 treatment.

### 2.4. Effects of CG200745 on Kidney Fibrosis in DSH Rats

We next performed Masson’s trichrome staining to investigate the efficacy of CG200745 as a potential therapeutic agent for renal fibrosis. As shown in Figure 2, the deposition of interstitial collagen was observed in the kidneys of DSH rats, which was attenuated by CG200745 treatment. Immunohistochemical staining for type IV collagen demonstrated an increased accumulation of type IV collagen in the peritubular and periglomerular interstitium in the kidneys of DSH rats, which was less pronounced in rats that received CG200745 (Figure 2). Furthermore, we investigated the effects of CG200745 on the expression of the myofibroblast molecular markers αSMA and fibronectin. In the kidneys of DSH rats, levels of αSMA and fibronectin increased, and this effect was prevented by CG200745 treatment (Figure 3a). Immunohistochemical staining for αSMA revealed an increased expression of αSMA in the peritubular and periglomerular interstitium in the kidneys of DSH rats, which was significantly reduced by CG200745 treatment (Figure 3b). We also investigated mRNA expression levels of genes encoding αSMA, fibronectin, and collagen I. As shown in Figure 4a, DSH rats presented with significantly increased levels of renal αSMA, fibronectin, and collagen I. All of these changes were attenuated in rats by the co-administration of CG200745.

### 2.5. Effects of CG200745 on TGF-β–Smad Signaling in DSH Rats

We next investigated whether CG200745 affects TGF-β/Smad signaling, which is known to be a critical mediator of renal fibrosis. Western blot analysis showed that protein expression levels of TGF-β1 and phosphorylated Smad2/3 were significantly increased in the kidneys of DSH rats (Figure 5a). Immunohistochemical staining also confirmed the increased expression of TGFβ1 in the kidneys of DSH rats, which was suppressed by CG200745 treatment (Figure 5b). Additionally, Smad4 levels were increased in DSH rats compared to those in control animals, which was counteracted by CG200745 treatment. We also investigated *TGFβ1* mRNA expression levels and found that these were increased in DSH rats (Figure 5a). This increase was prevented by treatment with CG200745.

### 2.6. Effects of CG200745 on Inflammatory Markers in DSH Rats

To evaluate DSH-induced inflammation and oxidative stress, we measured ED-1 and HO-1 levels in kidney tissues. We found that the increased expression of ED-1 and HO-1 in the kidneys of DSH rats was prevented by CG200745 treatment (Figure 6a). Immunohistochemical staining for ED-1, a marker of the murine macrophage population, confirmed that this was increased in the kidneys of DSH rats, which was accordingly suppressed by treatment with CG200745 (Figure 6b). We also investigated the expression of TNFα, which is a key inflammatory cytokine produced by infiltrating cells. As shown in Figure 4b, DSH significantly induced renal *TNFα* mRNA expression, and this increase was not observed in rats treated with CG200745. Increased expression levels of chemokines and adhesion molecules such as MCP-1, ICAM-1, and VCAM-1, which can activate, recruit, or induce the transmigration of inflammatory cells into the site of kidney injury, were also detected in DSH rats. Notably, co-treatment with CG200745 significantly reduced the expression of these chemokines in the kidneys of DSH rats (Figure 4b).

### 2.7. Effects of CG200745 on Apoptosis Markers in DSH Rats

The ratios of cleaved caspase 3 expression to caspase 3 levels, as well as the Bax/Bcl2 ratio, were markedly increased in the kidneys of DSH rats (Figure 7a). Accordingly, the number of tubular epithelial cells containing TUNEL-positive nuclei increased in the kidneys of DSH rats, which was attenuated by co-treatment with CG200745 (Figure 7b).

### 2.8. Effects of CG200745 on Control Rats

We also treated control rats with CG200745 to evaluate its toxicity. Body weight (BW), left ventricle weight/BW, kidney weight/BW, and blood pressure were not different between the two groups (data not shown). Figure 8 shows that increased acetylation was observed in the CG200745-treated group compared to that in the control group. However, there was no difference between controls and control rats treated with CG200745 in terms of the protein expression of fibrosis, inflammation, and apoptosis markers. These results suggested that CG200745 was not toxic to the control group. This suggests the disease-specific, anti-fibrosis, anti-inflammatory, and anti-apoptosis effects of GC200745.

## 3. Discussion

In the present study, we showed that treatment with the recently developed pan-HDACi CG200745 can prevent renal injury in the DSH rat model. As expected, DSH rats showed systemic arterial hypertension and proteinuria, which was attenuated by CG200745 treatment. Moreover, CG200745 inhibited the TGF-β/Smad-dependent signaling pathway, which contributes to renal fibrosis induced by DSH. CG200745 treatment also reduced the levels of inflammatory cytokines and the extent of apoptosis, which contributes to renal damage caused by DSH (Figure 9).

CG200745 has been shown to inhibit the proliferation of colon cancer [8], prostate cancer [9], non-small cell lung cancer [10], pancreatic cancer [11], and cholangiocarcinoma [12] cell lines. Moreover, the anti-fibrotic and anti-hypertrophic effects of CG200745 have been recently reported in a cardiac hypertrophy model [13]. However, whereas HDACis have been shown to attenuate cardiac hypertrophy, cardiac fibrosis [14,15], and hypertension [2], the renoprotective effects of HDACis have not been reported with respect to DOCA-salt hypertensive nephropathy.

In the current study, CG200745 treatment reduced the expression levels of inflammatory markers. Inflammation has been demonstrated to participate in the initiation and progression of glomerulosclerosis and tubulointerstitial fibrosis [16]. Inflammation is also known as a key initiator of cardiac extracellular matrix remodeling as it promotes the infiltration of inflammatory cells and the activation of fibroblasts [17,18]. It has been previously reported that HDACis suppress inflammatory cytokine expression and nitric oxide release [19]. They were also shown to exhibit therapeutic effects in an animal model of inflammation [20]. The present study showed that CG200745 treatment can prevent the increase in expression of the macrophage infiltration marker ED-1, the oxidative stress marker HO-1, and inflammatory cytokines and chemokines in DSH rats. These findings suggest that CG 200745 prevents renal injury by suppressing inflammation.

The multifunctional cytokine TGFβ induces the differentiation of fibroblasts to myofibroblasts, and is thought to be a key cellular inducer of fibrotic diseases, including diabetic nephropathy [21] and hypertensive nephropathy [22]. TGFβ induces fibrogenesis by promoting smooth muscle cell proliferation, stimulating extracellular matrix protein synthesis, and inhibiting matrix degradation [23,24]. In the present study, we found that CG200745 downregulates fibronectin protein levels and the mRNA levels of genes encoding fibronectin, TGFβ, and α-SMA, which were elevated in DSH rats, indicating that this treatment might prevent progressive renal injury by suppressing renal fibrosis.

Apoptosis is increased during progressive kidney disease [25]. A higher number of apoptotic cells might reflect the increased number of cells associated with inflammatory reactions. Further, apoptosis is associated with upregulation of the pro-apoptotic protein Bax, downregulation of the anti-apoptotic protein Bcl-2, and increased caspase-3 activity [26,27,28]. We observed significantly increased levels of Bcl-2 in DSH rats treated with CG200745 as compared to those in untreated DSH rats. Furthermore, the number of TUNEL-positive cells was significantly lower in the kidneys from CG200745-treated DSH rats.

In conclusion, our findings suggest that CG200745 can effectively reverse the progression of renal injury in the DSH rat model by exerting anti-inflammatory, anti-fibrotic, and anti-apoptotic effects by attenuating ED-1 overexpression, downregulating TGFβ/Smad signaling, and reducing the Bax/Bcl-2 ratio in DSH rat kidneys.

## 4. Materials and Methods

### 4.1. Animals

The experimental protocol was approved by the Animal Care Regulations Committee of Chonnam National University Medical School (CNU IACUC-H-2017-42) on 20 June 2017. Male Sprague–Dawley rats weighing 180 to 200 g were used. DSH was induced by the subcutaneous implantation of silicone rubber containing DOCA (200 mg/kg) one week after unilateral nephrectomy. Physiologic saline was supplied as drinking water to all animals. Two weeks after DOCA implantation, DOCA-salt rats were randomly divided and administered (via drinking) physiologic saline with or without GG200745 (5 mg/kg/day) for two weeks. Systolic blood pressure was measured by the tail cuff method (Kent Scientific Corporation, Torrington, CT, USA). The rats were maintained individually in metabolic cages for the last three days to allow for urine collection to measure Na^+^, creatinine, and microalbumin, and to calculate the albumin-to-creatinine ratio.

The rats were euthanized for semiquantitative immunoblotting and immunohistochemical studies four weeks after DOCA implantation. Rats were anesthetized with isoflurane and blood was collected from the inferior vena cava and analyzed for creatinine and Na^+^. The right kidney was rapidly removed and processed for immunoblotting, described as follows.

Another series of experiments was performed for real-time polymerase chain reaction (PCR) assays. The rats were decapitated in a conscious state and their kidneys were removed and maintained at −70 °C until being assayed for mRNA expression by real time-PCR.

Another series of experiments was performed to evaluate the toxicity of CG200745. Male Sprague–Dawley rats weighing 180 to 200 g were used. One week after unilateral nephrectomy, physiologic saline was supplied as drinking water to all animals. Two weeks later, these rats were randomly divided into two groups administered physiologic saline with or without GG200745 (5 mg/kg/day) for two weeks; the next steps, such as western blotting and PCR, were performed as described previously herein.

4.2. mRNA Expression of Inflammatory and Fibrosis Markers

The renal cortex was homogenized in Trizol reagent (Invitrogen, Carlsbad, CA, USA). RNA was extracted with chloroform, precipitated with isopropanol, washed with 75% ethanol, and then dissolved in distilled water. The RNA concentration was determined by the absorbance reading at 260 nm (Ultraspec 2000; Pharmacia Biotech, Cambridge, UK). The mRNA expression of genes encoding inflammatory cytokines and adhesion molecules was determined by real-time PCR. cDNA was produced via reverse transcription using 5 µg of total RNA with oligo(dT) priming and superscript reverse transcriptase II (Invitrogen). cDNA was quantified using the Smart Cycler II System (Cepheid, Sunnyvale, CA, USA) and SYBR Green was used for detection. Each PCR reaction contained 10 mM forward primer, 10 mM reverse primer, 2× SYBR Green Premix Ex Taq (TAKARA BIO INC, Shiga, Japan), 0.5 mL cDNA, and H_2_O to bring the final volume to 20 μL. Relative levels of mRNA were determined by real-time PCR using a Rotor-GeneTM 3000 Detector System (Corbette research, Mortlake, Australia). Table 3 shows the primer sequences for real-time PCR. The comparative critical threshold (*C*_t_) values from quadruplicate measurements were used to calculate gene expression, with normalization to GAPDH as an internal control. Melting curve analysis was performed to enhance the specificity of the amplification reaction.

### 4.2. Semiquantitative Immunoblotting

The dissected whole kidneys were homogenized in ice-cold isolation solution containing 0.3 M sucrose, 25 mM imidazole, 1 mM ethylenediamine tetraacetic acid, 8.5 µM leupeptin, and 1 mM phenylmethylsulfonyl fluoride, at pH 7.2. The homogenates were centrifuged at 1,000× *g* for 15 min at 4 °C to remove whole cells, nuclei, and mitochondria. The total protein concentration was then measured (Pierce BCA protein assay reagent kit, Pierce, Rockford, IL, USA). All samples were adjusted to achieve the same final protein concentrations using isolation solution, solubilized at 65 °C for 15 min in SDS-containing sample buffer, and then stored at −20 °C. To confirm equal loading of protein, an initial gel was stained with Coomassie blue. Sodium dodecyl sulfate-polyacrylamide gel electrophoresis was performed on 9% or 12% polyacrylamide gels. The proteins were transferred by gel electrophoresis (Bio-Rad Mini Protean II, Bio-Rad, Hercules, CA, USA) onto nitrocellulose membranes (Hybond ECL RPN3032D, Amersham Pharmacia Biotech, Little Chalfont, UK). The blots were subsequently blocked with 5% milk in PBST (80 mM Na_2_HPO_4_, 20 mM NaH_2_PO_4_, 100 mM NaCl, 0.1% tween 20, pH 7.5) for 1 h and incubated overnight at 4 °C with primary antibodies (TGFβ1 and fibronectin, Santa Cruz Biotechnology, Santa Cruz, CA, USA; ED-1, Serotec; α-smooth muscle actin (αSMA), Sigma Chemical, St. Louis, MO, USA; Bax and Bcl2, Cell Signaling Technology, Danvers, MA, USA), which was followed by incubation with secondary anti-rabbit (P448, DAKO, Glostrup, Denmark) or anti-mouse (P447, DAKO) horseradish peroxidase-conjugated antibodies. The labeling was visualized using an enhanced chemiluminescence system.

### 4.3. Immunohistochemistry

Kidney tissues were fixed with 4% paraformaldehyde, embedded in paraffin, and cut into 2-μm-thick sections. Hematoxylin and eosin staining was performed to assess histological morphology. The kidney tissue section slides were incubated in Gill’s hematoxylin for 5 min, washed with tap water, incubated in 95% ethanol, and stained with eosin and phloxine for 1 min. Subsequently, the sections were dehydrated in ethanol and xylene and were mounted with Canada balsam. For Masson’s trichrome staining, after deparaffinization with xylene, the sections were treated with Bouin’s solution at 56 °C for 30 min and were washed under running tap water until the sections were clear. The sections were subsequently stained with Weigert’s hematoxylin (A:B = 1:1), followed by staining with Biebrich Scarlet/Acid Fuchsin solution for 10 min and subsequent washing with distilled water. The sections were incubated with a phosphotungstic acid/phosphomolybdic acid solution for 10 min and treated with Aniline Blue solution for 15 min. They were subsequently incubated with acetic acid for 1 min and dehydrated with ethanol and xylene. Collagen deposition, nuclei, and muscle fibers were stained blue, black, and red, respectively. Immunoperoxidase labeling was performed as previously described [29].

### 4.4. Terminal Deoxynucleotidyl Transferase-Mediated dUTP Nick-End Labeling Assay

The ApopTag in situ apoptosis detection kit (Oncor, Gaithersburg, MD, USA) was used. The sections were dewaxed and treated with proteinase K and then incubated with equilibration buffer for 10 min, which was followed by incubation with working-strength TdT enzyme solution at 37 °C for 2 h. The reaction was terminated by incubating the samples in working-strength stop/wash buffer for 30 min at 37 °C. Sections were incubated with anti-digoxigenin peroxidase and then with diaminobenzidine and 0.01% H_2_O_2_ for 5 min at room temperature. The sections were counterstained with hematoxylin and examined by light microscopy [30].

### 4.5. Statistical Analyses

Results are expressed as the mean ± SE. Multiple comparisons among the groups were made by one-way ANOVA and post-hoc Tukey HSD tests. Differences with values of *p* < 0.05 were considered significant.

## Figures and Tables

**Figure 1 ijms-20-00508-f001:**
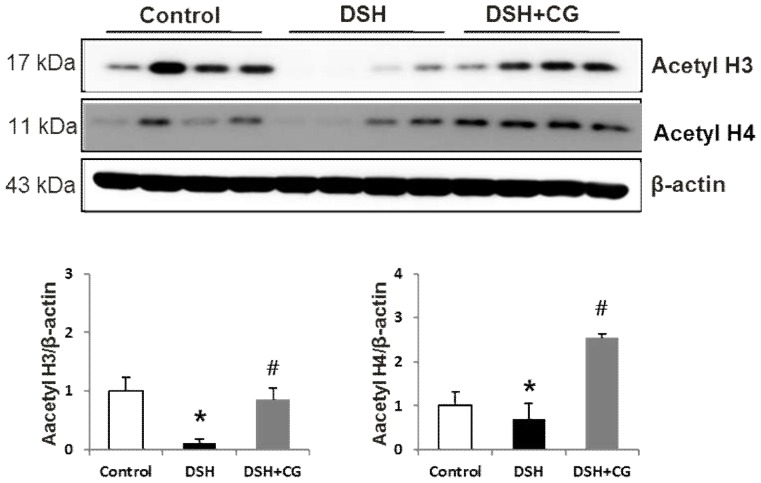
Semiquantitative immunoblotting analysis of acetyl H3 and acetyl H4 in the kidneys of experimental rats. Densitometric analysis revealed that the protein expression of acetyl H3 and acetyl H4 was decreased in deoxycorticosterone acetate (DOCA)-salt hypertensive rats (DSH) compared to that in controls, which was counteracted by CG200745 treatment (DSH + CG). * *p* < 0.05 compared to control. ^#^
*p* < 0.05 compared to DSH.

**Figure 2 ijms-20-00508-f002:**
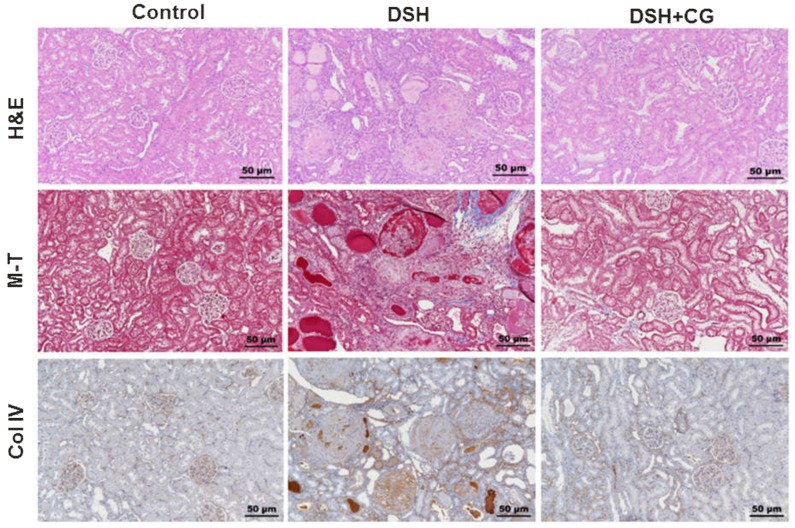
Hematoxylin and Eosin (H&E) stain, Masson’s trichrome (M-T), and collagen IV staining in the kidney cortex of experimental animals. Increased glomerulosclerosis and interstitial fibrosis were observed in deoxycorticosterone acetate (DOCA)-salt hypertensive (DSH) rats, which were attenuated by CG200745 treatment (DSH + CG).

**Figure 3 ijms-20-00508-f003:**
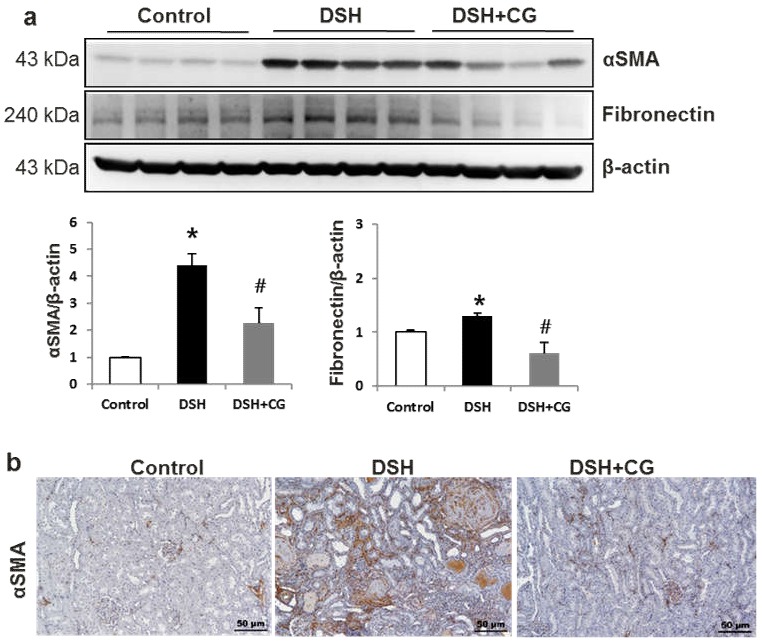
CG2007745 treatment suppresses the increase in myofibroblast markers observed in the kidneys of deoxycorticosterone acetate (DOCA)-salt hypertensive (DSH) rats. (**a**) Semiquantitative immunoblotting analysis of αSMA and fibronectin in the kidney. Densitometric analysis revealed that the protein expression of αSMA and fibronectin was increased in DSH rats compared to that in controls, which was counteracted by CG200745 treatment (DSH + CG); * *p* < 0.05 compared to controls, ^#^
*p* < 0.05 compared to DSH group. (**b**) Immunoperoxidase microscopy of αSMA in the cortex. Increased immunolabeling was observed in DSH rats, which was reversed by CG200745 treatment. Magnification: 100×.

**Figure 4 ijms-20-00508-f004:**
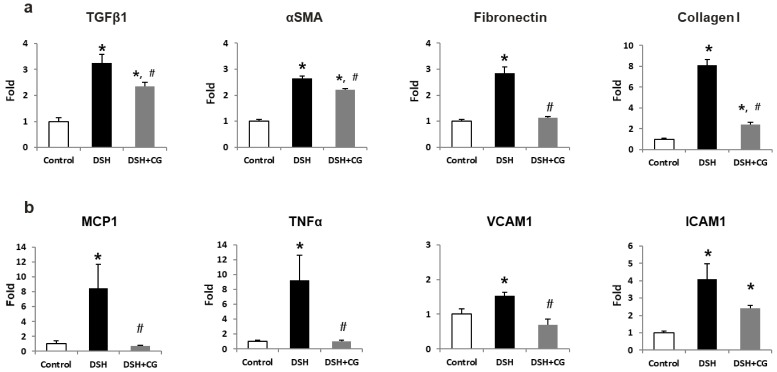
mRNA expression of fibrosis (**a**) and inflammatory markers (**b**) in the kidneys of deoxycorticosterone acetate (DOCA)-salt hypertensive (DSH) rats and those treated with CG200745 (DSH + CG). Columns show real time PCR data. * *p* < 0.05 compared to controls. ^#^
*p* < 0.05 compared to DSH rats.

**Figure 5 ijms-20-00508-f005:**
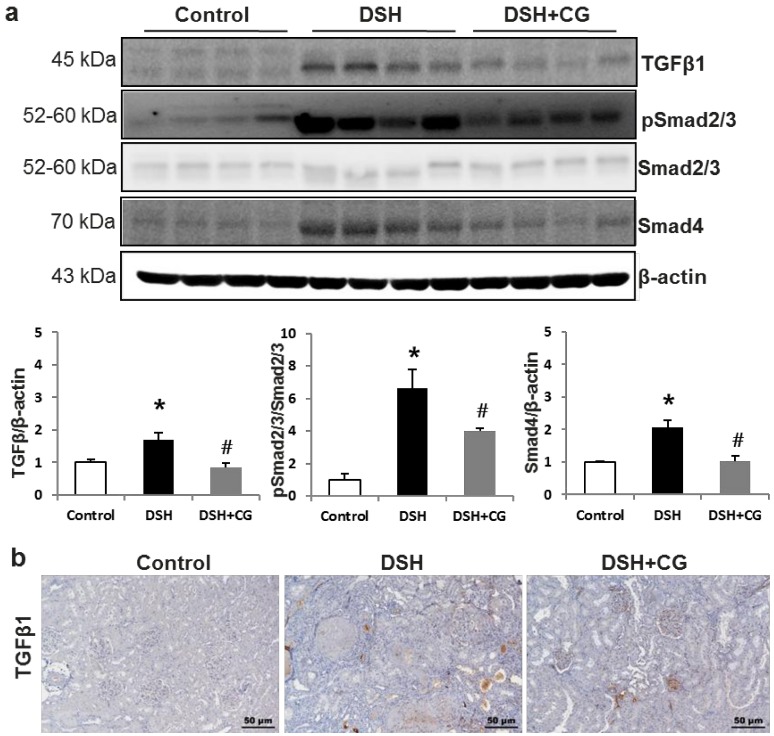
CG200745 suppresses the observed increase in TGF-β–Smad signaling observed in deoxycorticosterone acetate (DOCA)-salt hypertensive (DSH) rats. (**a**) Semiquantitative immunoblotting of TGFβ1 and Smad in the kidney. Densitometric analysis revealed that the protein expression of phospho-Smad 2/3 and Smad4was increased in DSH rats as compared to that in controls, which was counteracted by CG200745 treatment; * *p* < 0.05, when DHS or DSH + CG groups were compared to control group. *^#^ p* < 0.05, when DSH + CG group was compared to DSH group. (**b**) Immunoperoxidase microscopy of TGFβ1 in the kidney cortex. Increased immunolabeling was observed in the DSH group, which was counteracted by CG200745 treatment. Magnification: 100×.

**Figure 6 ijms-20-00508-f006:**
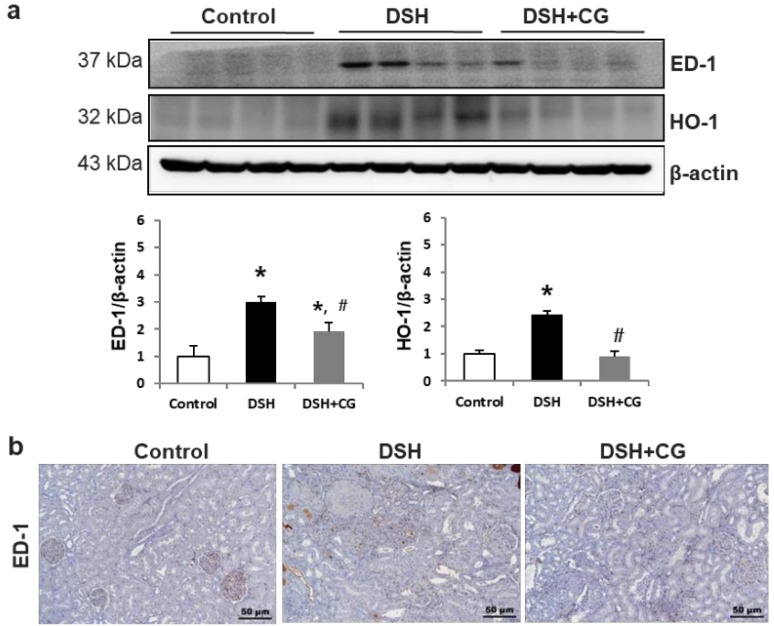
CG200745 suppresses inflammatory markers in deoxycorticosterone acetate (DOCA)-salt hypertensive (DSH) rats. (**a**) Semiquantitative immunoblotting for ED-1 and HO-1 in the kidney. Densitometric analysis revealed that the protein expression of ED-1 and HO-1 was increased in DSH rats, which was counteracted by CG200745 (DSH + CG); * *p* < 0.05, when DHS or DSH + CG groups were compared to control group. *^#^ p* < 0.05, when DSH + CG group was compared to DSH group. (**b**) Immunoperoxidase microscopy of ED-1 in the cortex. Increased immunolabeling was observed in DSH rats, which was reversed by CG200745 treatment. Magnification: 100×.

**Figure 7 ijms-20-00508-f007:**
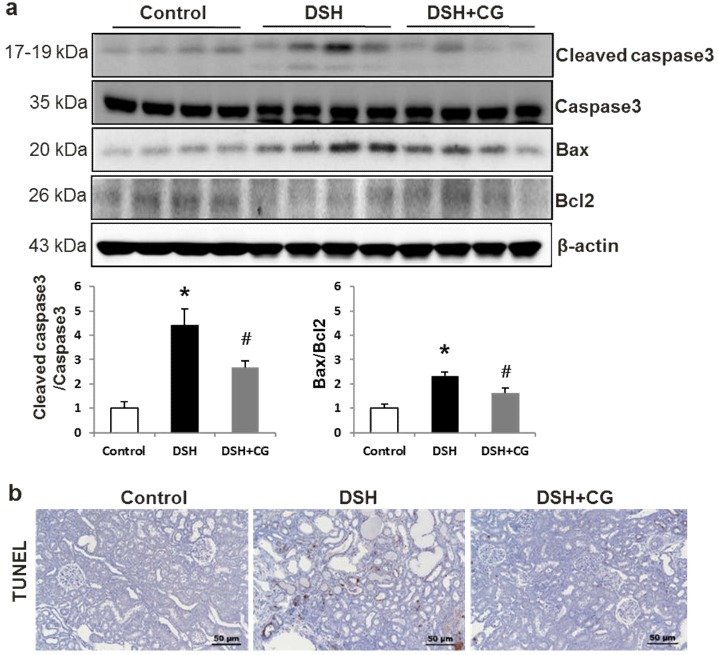
CG200745 deceases apoptosis in the kidneys of deoxycorticosterone acetate (DOCA)-salt hypertensive (DSH) rats. (**a**) The expression of cleaved caspase3 was increased, which was attenuated by CG200745 treatment (DSH + CG). The Bax/Bcl2 ratio was also markedly increased in DSH rats, which was attenuated by CG200745 treatment. The expression of Bax was increased, whereas Bcl2 was decreased in the DSH groups compared to those in controls. The Bax/Bcl2 ratio was markedly increased in DSH rats, which was attenuated by CG200745 treatment; * *p* < 0.05, when DHS or DSH + CG groups were compared to control group. *^#^ p* < 0.05, when DSH + CG group was compared to DSH group. (**b**) TUNEL assays revealed increased renal tubular apoptosis in the DSH group, whereas CG200745 co-treatment reduced the number of TUNEL-positive cells. Magnification: 100×.

**Figure 8 ijms-20-00508-f008:**
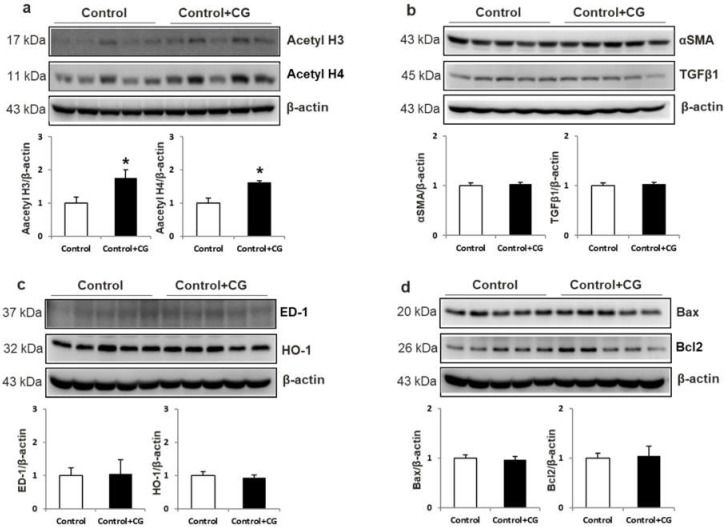
Toxicity of GC200745 (CG) to control rats. Semiquantitative immunoblotting of acetyl H3 and acetyl H4 in the kidney. Densitometric analysis revealed that the protein expression of acetyl H3 and acetyl H4 was increased in the CG-treated control group compared to that in controls (**a**). α-SMA and TGF-β (**b**), ED-1 and HO-1 (**c**), and Bax and Bcl2 (**d**) expression levels were not different between control and control + CG groups. * *p* < 0.05, when control + CG groups were compared to control group.

**Figure 9 ijms-20-00508-f009:**
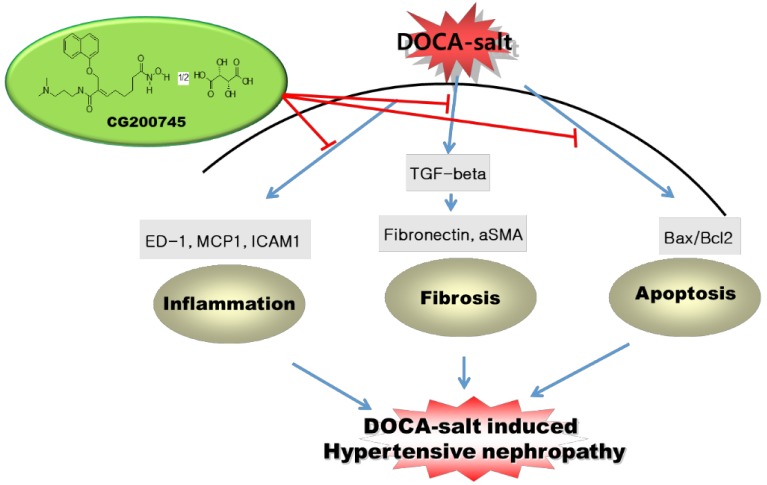
Schematic representation of the main findings of the study.

**Table 1 ijms-20-00508-t001:** Blood pressure and organ weights among experimental groups of rats.

Parameter	Control	DSH	DSH + CG200745
Body weight (g)	387.5 ± 3.33	257.5 ± 22.13 *	280.0 ± 12.25 *
Kidney weight (g)/BW (kg)	5.2 ± 0.3	14.2 ± 1.33 *	11.3 ± 0.8 *
LV weight(g)/BW (kg)	2.0 ± 0.05	3.9 ± 0.14 *	3.3 ± 0.17 *^,#^
SBP (mmHg)	119.0 ± 3.0	207.8 ± 9.7 *	171.7 ± 9.8 *^,#^
Urine output (mL/24 h)	21 ± 6.52	86.9 ± 16.46 *	87.0 ± 16.97 *

Values are expressed as mean ± SE. These values were measured on the last day of experiments (week 4). DSH, deoxycorticosterone acetate salt hypertensive rats; SBP, systolic blood pressure; BW, body weight; LV, left ventricle. * *p* < 0.05, when DHS or DSH + CG200745 groups were compared to control group. *^#^ p* < 0.05, when DSH + CG200745 group was compared to DSH group.

**Table 2 ijms-20-00508-t002:** Changes in renal function among experimental groups of rats.

Parameter	Control	DSH	DSH + CG200745
BUN	14.8 ± 0.4	21.2 ± 5.36	19.7 ± 3.41
Creatinine	0.24 ± 0.05	0.31 ± 0.08	0.25 ± 0.03
Serum Na	120.0 ± 4.5	118.3 ± 5.5	125.5 ± 3.4
Urine Na	303.0 ± 11.5	159.3 ± 10.64 *	169.5 ± 9.70 *
FE_Na_	2.4 ± 0.67	12.3 ± 3.07 *	9.9 ± 2.68
ACR (mg/gCr)	297.1 ± 197.3	5864.0 ± 662.2 *	3681.1 ± 267.3 *^,#^

Values are expressed as the mean ± SE. These values were measured on the last day of experiments. BUN, blood urea nitrogen, P-Na, plasma sodium; FE_Na_, fractional excretion of sodium into urine; ACR, albumin-to-creatinine ratio; Cr, Creatinine; DSH, deoxycorticosterone acetate salt hypertensive rats. * *p* < 0.05, when DHS or DSH + CG200745 groups were compared to control group. *^#^ p* < 0.05, when DSH + CG200745 group was compared to DSH group.

**Table 3 ijms-20-00508-t003:** Primer sequences for real-time PCR.

Primers	Sequence
TGFβ1	Sense: GGACTACTACGCCAAAGAAG Antisense: TCAAAAGACAGCCACTCAGG
αSMA	Sense: TGTGCTGGACTCTGGAGATG Antisense: GAAGGAATAGCCACGCTCAG
Fibronectin	Sense: CATGAAGGGGGTCAGTCCTA Antisense: GTCCATTCCCCTTTTCCATT
Collagen I	Sense: CAACCTCAAGAAGTCCCTGC Antisense: ACAAGCGTGCTGTAGGTGAA
MCP1	Sense: CTGCTACTCATTCACTGGC Antisense: CTTCTGGACCCATTCCTTAT
TNFα	Sense: GTCGTAGCAAACCACCAAGC Antisense: CTCCTGGTATGAAATGGCAA
VCAM1	Sense: GGGGGCCAAGTCCGTTCTGA Antisense: GGGGGCCACTGAATTGAATC
ICAM1	Sense: AAGGTGTGATATCCGGTAGA Antisense: CCTTCTAAGTGGTTGGAACA
GAPDH	Sense: ATCAAATGGGGTGATGCTGGTGCTG Antisense: CAGGTTTCTCCAGGCGGCATGTCAG

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
