# Peer review of "Renoprotective Effect of the Histone Deacetylase Inhibitor CG200745 in DOCA-Salt Hypertensive Rats"

_ijms, 2019, doi:10.3390/ijms20030508_

Reviewer 1 Report

Previously, histone deacetylase inhibitor GC200745 drug has been showed to be beneficial on renal fibrosis in obstructive kidney disease, hypertension and hyperglycemia in Cushing's syndrome, ameliorating pulmonary arterial hypertension induced by ROS, attenuates cardiac hypertrophy in DOCA-induced hypertensive rats. The present study by Bae E et al examined  the  renoprotective effect of GC200745. Authors has used DOCA salt hypertensive rats and compared DOCA salt and DOCA salt + GC200745 treated rats. DOCA salt treated animals has increase in collagen, fibronectin, MCP1, TNF-a, kidney fibrosis etc.,  and GC200745 drug almost revert those effects. Overall, GC200745 revert  the renal injury by acting as anti-inflammatory, anti-fibrotic, and anti-apoptotic effects in DOCA salt induced hypertensive model. It is appreciated that author exploited multiple approach to provide sufficient evidence on renal effect of GC200745. 

Comments

Both obstructive kidney      disease and DOCA salt induces inflammatory cytokines, increase the      apoptosis, and increase in fibrosis in similar manner. Even though this      study explored renoprotective effect of GC200745 in DOCA salt induced      fibrosis and apoptosis, similar effect of  inflammatory, apoptosis, fibrotic pathway      shared.  That questions the      originality/novelty of this study.

Conundrum      of GC200745 is that it has different action or opposite effect in      different cells. In cancer cells it      induces cell cycle arrest, differentiation, and apoptosis while in renal      system it seems to reduces apoptosis and inflammation. What is mechanism      behind action of GC200745 is different in two different system? Does      increased p53 acetylation by HDAC inhibitors in renal system decrease the      apoptosis to protect the kidney? It would be interesting to see what is      the effect of GC200745 in control rats?

Author mentioned renoprotective      effects of HDACi have not been reported so far, while there is study      showing CG200745 attenuates renal fibrosis in obstructive kidney disease      from the same group.

Why animals were given      DOCA or GC200745 with saline? Why did animals undergone unilateral      nephrectomized? I mean why renal insufficiency is necessary ?

In our experience and also      based on the literature, DOCA salt does not change the body weight. Here      author showed DOCA salt reduce the body weight and inhibitor GC200745      partially restores the BW. Does this drug has an effect on weight gain? At      the same time GC200745 increase kidney and LV weight significantly. It is      confusing in terms of mechanism.

Does this drug induces NO      production or suppress it?

7.     Why did author used tail-cuff method? Even though it is convenient and cost effective to use but there  is question about the reliability of tail-cuff method.

8.     Add limitation of the study because GC200745 is not well validated such as side effect of these compound, inhibiting/activating any other metabolic pathways, etc.

Author Response

Dear reviewer,

We are grateful for the very careful and detailed consideration that you have provided regarding our manuscript. We have revised the manuscript extensively in accordance with your specific concerns that arose from the Editorial Conference, as well as those raised by the reviewers. We have made every attempt to thoroughly address these comments. Please find our revised manuscript enclosed, as well as our itemized responses. We hope that you will now find our manuscript suitable for publication.

Thank you and we look forward to hearing your response.

Best regards,

Soo Wan Kim & Eun Hui Bae

##Response to Reviewer 1 Comments,

Previously, histone deacetylase inhibitor GC200745 drug has been showed to be beneficial on renal fibrosis in obstructive kidney disease, hypertension and hyperglycemia in Cushing's syndrome, ameliorating pulmonary arterial hypertension induced by ROS, attenuates cardiac hypertrophy in DOCA-induced hypertensive rats. The present study by Bae E et al examined the renoprotective effect of GC200745. Authors has used DOCA salt hypertensive rats and compared DOCA salt and DOCA salt + GC200745 treated rats. DOCA salt treated animals has increase in collagen, fibronectin, MCP1, TNF-a, kidney fibrosis etc., and GC200745 drug almost revert those effects. Overall, GC200745 revert the renal injury by acting as anti-inflammatory, anti-fibrotic, and anti-apoptotic effects in DOCA salt induced hypertensive model. It is appreciated that author exploited multiple approach to provide sufficient evidence on renal effect of GC200745. 

Point 1: Both obstructive kidney disease and DOCA salt induces inflammatory cytokines, increase the apoptosis, and increase in fibrosis in similar manner. Even though this study explored renoprotective effect of GC200745 in DOCA salt induced fibrosis and apoptosis, similar effect of inflammatory, apoptosis, fibrotic pathway shared. That questions the originality/novelty of this study.

Response 1: We agree with the reviewer’s comment. However, DOCA salt hypertension and UUO are models that cause kidney damage due to different clinical mechanisms. The pathophysiological findings of the UUO model differ significantly from those observed during hypertensive nephrosclerosis. We thus think that it is meaningful to evaluate the effects of CG200745 using each model.

Point 2: Conundrum of GC200745 is that it has different action or opposite effect in different cells. In cancer cells it induces cell cycle arrest, differentiation, and apoptosis while in renal system it seems to reduces apoptosis and inflammation. What is mechanism behind action of GC200745 is different in two different system? Does increased p53 acetylation by HDAC inhibitors in renal system decrease the apoptosis to protect the kidney? It would be interesting to see what is the effect of GC200745 in control rats?

Response 2: We agree with the reviewer. Accordingly, we conducted additional experiments using control animals and included these data in Figure 8. In the control group, CG200745 did not result in any observed physiological changes, which suggests that the effects of CG200745 might be disease-specific.

Point 3: Author mentioned renoprotective effects of HDACi have not been reported so far, while there is study showing CG200745 attenuates renal fibrosis in obstructive kidney disease from the same group.

Response 3: We appreciate the reviewer’s comment. We corrected this to be “the renoprotective effects of HDACis have not been reported with respect to DOCA-salt hypertensive nephropathy”.

Point 4: Why animals were given DOCA or GC200745 with saline? Why did animals undergone unilateral nephrectomized? I mean why renal insufficiency is necessary?

Response 4: We followed a previously published method to generate the DOCA-salt hypertensive model (J Am Soc Nephrol. 1999 Jan;10 Suppl 11:S143-8) This method is conducted as follows: 1 week before nephrectomy and 1 week later, DOCA should be injected subcutaneously and 0.9% or 1% saline should be supplied as drinking water. We do not know the exact reason for unilateral nephrectomy, but it might increase renin–angiotensin levels and aggravate hypertension (Acta Pathol Microbiol Scand A. 1970;78(6):669-73). These types of experiments in rats lead to a similar degree of hypertension, which most closely resembles clinically-occurring secondary hypertension such as primary hyperaldosteronism.

Point 5: In our experience and also based on the literature, DOCA salt does not change the body weight. Here author showed DOCA salt reduce the body weight and inhibitor GC200745 partially restores the BW. Does this drug has an effect on weight gain? At the same time GC200745 increase kidney and LV weight significantly. It is confusing in terms of mechanism.

Response 5: Previously, many papers have shown that DOCA salt treatment decrease body weight (BW) and increases the relative left ventricle (LV) weight/BW and kidney weight/BW (Kee HJ et al. Kidney Blood Press Res. 2013;37(4-5):229-39; Bae EH et al. Kidney Blood Press Res. 2012;36(1):248-57; Bae EH et al. Nephrol Dial Transplant. 2010 Apr;25(4):1051-9). The relative increase in LV weight/BW and kidney weight/BW was attenuated when CG200745 was administered.

Point 6: Does this drug induces NO production or suppress it?

Response 6: We appreciate this question and completely agree that it would be interesting to investigate NO levels in DOCA-salt hypertensive rats treated with CG200745. However, this paper focused on clarifying the renoprotective effects of CG200745 during DOCA salt hypertension. Thus, a comprehensive study to investigate the effect of CG200745 on NO/cGMP will be conducted separately within the next year.

Point 7: Why did author used tail-cuff method? Even though it is convenient and cost effective to use but there is question about the reliability of tail-cuff method.

Response 7: Direct measurements of blood pressure cannot rule out the effects of anesthetic agents on the hemodynamics. We have published many papers using the tail-cuff method. The CODA noninvasive BP system (a tail-cuff Method, Kent Scientific Corporation) was also published (Methods Mol Biol 2017;1614:69-73). We believe that the tail-cuff method is reliable.

Point 8: Add limitation of the study because GC200745 is not well validated such as side effect of these compound, inhibiting/activating any other metabolic pathways, etc.

Response 8: We conducted additional experiments using control animals and included these data in Figure 8. In the control group, CG200745 resulted in no side effects. Moreover, the inhibition/activation of metabolic pathways with CG200745 has also been published (Jung DE et al. Sci Rep. 2017;7(1):10921; Kim KP et al. Invest New Drugs. 2015;33(5):1048-57: Chun SM et al. PLoS One. 2015;10(3):e0119379; Hwang JJ et al. Invest New Drugs. 2012;30(4):1434-42; Oh ET et al. Invest New Drugs. 2012;30(2):435-42.)

Reviewer 2 Report

This is a very interesting study about probably a novel class/representative of nephroprotective agents that can be given orally. As a continuation of a previous study on the role of CG200745 in obstructive renal failure presented by Choi et al. (Histone deacetylase inhibitor, CG200745 attenuates renal fibrosis in obstructive kidney disease. Sci Rep. 2018 Aug 1;8(1):11546. doi: 10.1038/s41598-018-30008-5) this manuscript provides more information about mechanisms of tested drug renoprotection. Interestingly, authors present results of in vivo observations + histopathological/biochemical/molecular results and compare them in understandable way. I have only 2 requests: 1) please check carefully manuscript once again (line 115 written twice "was was"); 2) please focus on discussion chapter (line 197: the sentence "CG200745 exerted, attenuating renal fibrosis, inflammation and apoptosis" need to be corrected).

Author Response

Dear reviewer,

We are grateful for the very careful and detailed consideration that you have provided regarding our manuscript. We have revised the manuscript extensively in accordance with your specific concerns that arose from the Editorial Conference, as well as those raised by the reviewers. We have made every attempt to thoroughly address these comments. Please find our revised manuscript enclosed, as well as our itemized responses. We hope that you will now find our manuscript suitable for publication.

Thank you and we look forward to hearing your response.

Best regards,

Soo Wan Kim & Eun Hui Bae

##Response to Reviewer 2 Comments,

This is a very interesting study about probably a novel class/representative of nephroprotective agents that can be given orally. As a continuation of a previous study on the role of CG200745 in obstructive renal failure presented by Choi et al. (Histone deacetylase inhibitor, CG200745 attenuates renal fibrosis in obstructive kidney disease. Sci Rep. 2018 Aug 1;8(1):11546. doi: 10.1038/s41598-018-30008-5) this manuscript provides more information about mechanisms of tested drug renoprotection. Interestingly, authors present results of in vivo observations + histopathological/biochemical/molecular results and compare them in understandable way. I have only 2 requests:

Point 1: please check carefully manuscript once again (line 115 written twice "was was");

Response 1: We apologize for this oversight, and we have corrected this.

Point 2: please focus on discussion chapter (line 197: the sentence "CG200745 exerted, attenuating renal fibrosis, inflammation and apoptosis" need to be corrected).
Response 2: Thank you for pointing this out. We revised the discussion section accordingly and also specifically corrected that sentence.

Reviewer 3 Report

The authors described that pan-HDAC inhibitor CG200745 prevented the progression of renal injury in DSH rats by exerting anti-inflammatory, anti-fibrotic, and anti-apoptotic effects. However, the work is not accepted.

1, Whether CG200745 regulated the hypertension in DSH model?

2, We found that the paper Antifibrotic activity of an inhibitor of histone deacetylases in DOCA-salt hypertensive rats published in Br J Pharmacol. The inhibitor of histone deacetylases has been reported to inhibit the progression of renal injury by anti-fibrotic activity. The authors only change another inhibitor to study and there was no significant.

3, The author should add a group of only CG200745 in animal model to explore the toxicity.

4, What the selectivity of Pan-HDAC inhibitor in vivo and in vitro?

5, The author should add the scale range in exhibited Figures (HEmasson, etc)?

6, The TUNEL figure in DSH group was wrong.

Author Response

Dear reviewer,

We are grateful for the very careful and detailed consideration that you have provided regarding our manuscript. We have revised the manuscript extensively in accordance with your specific concerns that arose from the Editorial Conference, as well as those raised by the reviewers. We have made every attempt to thoroughly address these comments. Please find our revised manuscript enclosed, as well as our itemized responses. We hope that you will now find our manuscript suitable for publication.

Thank you and we look forward to hearing your response.

Best regards,

Soo Wan Kim & Eun Hui Bae

##Response to Reviewer 3 Comments,

The authors described that pan-HDAC inhibitor CG200745 prevented the progression of renal injury in DSH rats by exerting anti-inflammatory, anti-fibrotic, and anti-apoptotic effects. However, the work is not accepted.

Point 1: Whether CG200745 regulated the hypertension in DSH model?

Response 1: More research is needed to determine whether CG200745 regulates hypertension in the DSH model. However, the present study showed that when CG200745 was administered to the sham group, there were no changes in blood pressure (as compared to that in the untreated sham group). This suggests that these effects should be considered a secondary decrease in blood pressure due to kidney protection in DSH rats mediated by the anti-inflammatory, anti-fibrotic, and anti-apoptotic effects of CG200745.

Point 2: We found that the paper “Antifibrotic activity of an inhibitor of histone deacetylases in DOCA-salt hypertensive rats” published in Br J Pharmacol. The inhibitor of histone deacetylases has been reported to inhibit the progression of renal injury by anti-fibrotic activity. The authors only change another inhibitor to study and there was no significant.

Response 2: GC200745 is a novel HDAC inhibitor and its effects on the kidneys of DOCA-salt hypertensive rats have not been studied. Therefore, we considered this study to be novel.

Point 3: The author should add a group of only CG200745 in animal model to explore the toxicity.

Response 3: In accordance with this suggestion, we conducted additional experiments to explore the toxicity of CG200745. We included these data as Figure 8.

Point 4: What the selectivity of Pan-HDAC inhibitor in vivo and in vitro?

Response 4: We obtained the target selectivity data from the manufacturer of this agent as follows. CG200745 shows stronger pan-HDAC inhibitory effects compared to those of Zolinza or PXD-101.

Point 5: The author should add the scale range in exhibited Figures (HEmasson, etc)?

Response 5: In accordance with this suggestion, we added the scale range to the figures.

Point 6: The TUNEL figure in DSH group was wrong.

Response 6: We revised this figure accordingly.

Round  2

Reviewer 1 Report

My concern was about toxicity of the drug. That seems to be no difference in BP on control animals. I am pleased with revision. 

Reviewer 3 Report

No comments